# Sudden Cardiac Death: The Role of Molecular Autopsy with Next-Generation Sequencing

**DOI:** 10.3390/diagnostics15040460

**Published:** 2025-02-13

**Authors:** Jennifer Fadoni, Agostinho Santos, António Amorim, Laura Cainé

**Affiliations:** 1National Institute of Legal Medicine and Forensic Sciences, North Branch, 4050-167 Porto, Portugal; jennifer.n.fadoni@inmlcf.mj.pt (J.F.);; 2LAQV&REQUIMTE, Laboratory of Applied Chemistry, Department of Chemical Sciences, Faculty of Pharmacy, University of Porto, 4050-313 Porto, Portugal; 3National Institute of Legal Medicine and Forensic Sciences, Centre Branch, 3000-548 Coimbra, Portugal; 4Faculty of Medicine, Porto University, 4200-319 Porto, Portugal

**Keywords:** sudden cardiac death, molecular autopsy, arrhythmias, cardiomyopathies, next-generation sequencing, NGS

## Abstract

Molecular autopsy is a term employed to describe the investigation of the cause of death through the analysis of genetic information using biological samples collected post-mortem. Its utility becomes evident in situations where conventional medico-legal autopsy methods are not able to identify the cause of death, i.e., in sudden cardiac death (SCD) cases in young individuals, where deaths are commonly due to genetic cardiac conditions, such as cardiomyopathies and channelopathies. The recent advancement in high-throughput sequencing techniques, such as next-generation sequencing (NGS), has allowed the investigation of a high number of genomic regions in a more cost-effective and faster approach. Unlike traditional sequencing methods, which can only sequence one DNA fragment at a time, NGS can sequence millions of short polynucleotide fragments simultaneously. This parallel approach reduces both the time and cost required to generate large-scale genomic data, making it a useful tool for applications ranging from basic research to molecular autopsy. In the forensic context, by enabling the examination of multiple genes or entire exomes and genomes, NGS enhances the accuracy and depth of genetic investigations, contributing to a better understanding of complex inherited diseases. However, challenges remain, such as the interpretation of variants of unknown significance (VUS), the need for standardized protocols, and the high demand for specialized bioinformatics expertise. Despite these challenges, NGS continues to offer significant promise for enhancing the precision of molecular autopsies. The goal of this review is to assess the effectiveness of contemporary advancements in molecular autopsy methodologies when applied to cases of SCD in young individuals and to present an overview of the steps involved in the analysis of NGS data and the interpretation of genetic variants.

## 1. Introduction

While the majority of cases of sudden cardiac death (SCD) occur in elderly subjects presenting coronary artery disease, in the young population, hereditary cardiac conditions exhibit a higher prevalence. Despite the lack of consensus on the definition of the term ‘young’, studies referring to SCD in young individuals typically analyze those aged up to 35 [1], 40 [2], or even 50 years old [3]. In these individuals, the causes of SCD can be grouped into (1) cardiomyopathies, characterized by structural abnormalities in the heart, and (2) channelopathies, in which, in the majority of cases, the heart maintains structural normality without discernible macroscopic or microscopic alterations [4]. The examination of deceased individuals who have experienced SCD at a young age often reveals structural heart abnormalities; however, approximately one-third of SCD cases in young individuals remain without a clear explanation even after a comprehensive post-mortem evaluation, which is commonly referred to as sudden arrhythmic death syndrome (SADS) or autopsy-negative sudden death [5]. Estimates of the annual incidence of SCD exhibit considerable variability, influenced by the applied definitions, data sources for case identification, and the methods employed in rate estimation. Nevertheless, estimates indicate that in Western countries, the incidence ranges from 50 to 100 per 100,000, with lower rates observed in Asian countries [6]. In such cases, the utilization of genetic testing conducted post-mortem, frequently termed a molecular autopsy, plays a vital role in identifying the underlying genetic cause of SCD [7]. A molecular autopsy is a term used for genetic testing from a deceased individual using blood or other tissue samples collected post-mortem. This process is employed to investigate potential underlying genetic factors that may play a significant role in causing death. The application of a genetic analysis in the investigation of genetic causes of SCD started more than two decades ago [8], with the conventional approach to molecular autopsy involving the sequencing of the exonic regions of selected genes using Sanger sequencing (first-generation sequencing) [9]. Despite its historical significance, Sanger sequencing has several limitations, including low throughput, high cost, and a larger quantity of input DNA. These constraints significantly hinder its utility in high-throughput and cost-efficient genomic analyses, especially in scenarios requiring extensive sequencing coverage or scalability. In the current molecular autopsy landscape, Sanger sequencing is used for confirming NGS findings of interest [10]. This utilization underscores the transformative impact of contemporary molecular biology on scientific investigation and our understanding of molecular pathogenesis across diverse medical fields. The development of molecular genetic tools has enabled a deeper understanding of the human genetic code, how this code varies among individuals, and the application of this knowledge in analyses tailored to specific targets. Moreover, recent developments in sequencing technologies, particularly NGS, have enabled comprehensive screening of substantial portions of the human genome in a more cost-efficient way [11]. Currently, there are several different NGS platforms employing a variety of sequencing technologies, although these platforms share an essential step where DNA templates amplified through clonal processes or individual molecules of DNA are sequenced in a massively parallel manner on a flow cell [12]. This multiplexed approach holds the capacity to significantly enhance the diagnostic rate of genetic testing by enabling simultaneous investigation of numerous genetic variants. Therefore, the substantial amount of data generated by NGS holds significant promise to enhance our comprehension of the genetic mechanisms involved in SCD. It establishes the foundation for precise diagnostics, prevents more cases of SCD by providing biomarkers to be investigated in first-degree relatives who might be at risk, and contributes to the development of targeted therapeutic strategies. In this review, we aim to examine the present accomplishments of molecular autopsy in young SCD cases, present a summary of the key steps involved in the analysis of NGS data and genetic variants, and address the limitations and challenges that need to be overcome in this field.

## 2. Materials and Methods

In November 2024, a search was conducted on the PubMed and Scopus databases using the terms ‘NGS’ OR ‘next-generation sequencing’ AND ‘molecular autopsy’ OR ‘post-mortem genetic testing’ AND ‘sudden cardiac death’ OR ‘sudden unexplained death’. Articles focusing on molecular autopsy and NGS in unexplained sudden cardiac death, sudden unexplained death syndrome, sudden arrhythmic death syndrome, or related inherited cardiac conditions in individuals of all age groups were considered. The present study is a narrative review; therefore, no formal criteria for study selection were employed.

## 3. Sudden Cardiac Death

Sudden cardiac death (SCD) is a devastating, abrupt, and unforeseen event that profoundly affects families and communities. Often, it is the first manifestation of an underlying pathology that had gone undetected and can be associated with numerous cardiovascular conditions. SCD is defined as a sudden and unforeseen death, presumably caused by a cardiac arrhythmia or a hemodynamic catastrophe, taking place within the first hour of symptom onset when witnessed or in cases where individuals are found deceased without having exhibited any symptoms in the previous 24 h [13]. It constitutes a significant public health challenge, responsible for nearly 50% of all cardiovascular-related deaths [6,13]. Estimates of the yearly incidence of SCD vary considerably, influenced by the definitions applied, data sources for case identification, and the methods used for estimating rates [14]. Furthermore, it encompasses differences in both the incidence and the causes of death observed across age groups. Specifically, approximately 19% of sudden natural deaths in children aged 1 to 13 years result from cardiac causes, while in the 14 to 21 years age group, the proportion increases to 30% [15,16].

### 3.1. Age Relation Risk of SCD

The occurrence of SCD is impacted by age, with the most rapid rise typically observed around 75 years of age [17]. The underlying causes of SCD diverge according to age groups (Figure 1). Coronary artery disease (CAD) is the major contributor to SCD cases in individuals over 35 years of age [18]. However, in individuals aged 35 years and younger, there is a significant shift in the distribution of causes, with a decline in the overall number of deaths attributable to CAD and a rise in SCD cases attributed to fatal complications of hereditary cardiac conditions, often referred to as inherited heart diseases [19]. These conditions encompass a group of cardiovascular disorders with a genetic basis, which can be categorized into cardiomyopathies and channelopathies [4].

#### 3.1.1. Cardiomyopathies

Cardiomyopathies result from pathological changes in proteins responsible for cardiac myocyte contractility and development, leading to the disruption of myocardial contraction ability, relaxation, and/or the viability of myocytes [21]. Three categories of cardiomyopathy exhibit robust genetic associations: hypertrophic cardiomyopathy (HCM), dilated cardiomyopathy (DCM), and arrhythmogenic cardiomyopathy (ACM) (Table 1).

#### 3.1.2. Channelopathies

Channelopathies arise from either acquired malfunctions of ion channels or genetic variants. Acquired channelopathies can be triggered by substances that have the potential to either inhibit or activate ion channels, such as immunoglobulins, toxins, or drug exposure. Genetic channelopathies can arise from genetic variants in genes responsible for encoding the subunit of ion channels (alpha subunit) or in genes that encode regulatory proteins, such as the beta subunit or enzymes responsible for regulating the activity of the alpha subunit [33]. The predominant phenotypes observed in individuals carrying genetic channelopathies are congenital long QT syndrome (LQTS), Brugada syndrome (BrS), catecholaminergic polymorphic ventricular tachycardia (CPVT), and short QT syndrome (SQTS) (Table 2).

## 4. Molecular Autopsy

After a case of SCD, efforts are made to determine the cause of death using premorbid clinical details and post-mortem pathological findings. In the current scenario, the majority of molecular autopsy studies follow the guidelines established by the International Heart Rhythm Society USA/European Heart Rhythm Association (HRS/EHRA) [43]. These guidelines suggest conducting either a comprehensive or targeted molecular autopsy in cases of SADS when it is suspected that the death resulted from a cardiomyopathy or channelopathy. The key aspects of the post-mortem investigation in SCD cases are summarized in Figure 2.

However, in up to 30% of SCD cases in young individuals, no cause of death is identified post-mortem, referred to as autopsy-negative or sudden arrhythmic death syndrome (SADS). In instances where the cause of death remains unknown, post-mortem genetic testing, also known as molecular autopsy, may identify a cause of death in up to 30% of SADS cases [44], and this determination of a genetic etiology is crucial not only for diagnosing and clarifying the most plausible cause of death but also for preventing SCD in the relatives of the victims, who might carry the same genetic variant. This highlights the critical role of molecular autopsy in SCD victims.

The term molecular autopsy, also known as post-mortem genetic testing, refers to the use of genetic studies on biological samples collected post-mortem as part of forensic examination, aimed at determining the genetic basis of the cause of death [45]. The first molecular autopsy was performed in 1999 by Ackerman et al. using Sanger sequencing, a method that represents a first-generation sequencing technology [8]. This method relies on a DNA polymerase reaction that synthesizes multiple copies of the desired DNA sequence [9]. For nearly three decades, Sanger sequencing has been the standard method for nucleic acid sequencing, demonstrating a high level of accuracy in identifying genetic variants [46]. Although Sanger sequencing is characterized by its simplicity and accuracy, it is considered a low-throughput technique, allowing for the sequencing of one sequence of DNA at a time [47]. This sequencing method, initially pivotal in the early molecular autopsy studies, has played a crucial role in genetic analysis. However, it has restricted throughput, requires a large input of DNA, and has a high cost-effectiveness, particularly when sequencing multiple genes simultaneously or analyzing the same genomic region across numerous samples. The established guidelines [43] recommend the application of post-mortem genetic testing, which typically involves sequencing the protein-coding exons of genes associated with these cardiac conditions. Several genes have been linked to these cardiac conditions, and the analysis of multiple genes as well as entire exomes or genomes has become achievable due to advancements in DNA sequencing techniques, particularly next-generation sequencing.

### 4.1. Next-Generation Sequencing

As a response to the limitations of initial sequencing technologies, a high-throughput method for nucleic acid sequencing, known as next-generation sequencing (NGS), has emerged. Also known as massive parallel sequencing (MPS), NGS is rapid and cost-efficient, enabling the sequencing of the entire human genome within a few days using just 1000 ng of DNA as input [46,48]. Simultaneously examining millions of short polynucleotide fragments varying in length from 50 to 250 base pairs (bp) and known as ‘short reads’, enabling high-throughput sequencing. Each read is matched with the corresponding read from the opposite end of the fragment, generating ‘paired-end’ reads. Subsequently, alignment algorithms are utilized to align the polynucleotide fragments with a sequence of reference, a process that aims to reconstruct the original DNA sequence. Following this, specialized software is employed to identify discrepancies between the reads and the sequence of reference, which could indicate a variant of interest (Figure 3).

However, determining the clinical significance of such variants necessitates further investigation. The accuracy and reliability of NGS have been proved by extensive validation in comparison to Sanger sequencing, and an NGS test can be tailored to focus on selected genes, the whole exome, or even the whole genome. Gene panels are typically composed of genes that have previously been linked to a specific phenotype. This approach is designed to optimize sensitivity, specificity, and coverage for the chosen genes. The determination of which genes to incorporate into the panel is at the discretion of the individual laboratory, and its cost is variable depending on customization [49]. Whole-exome sequencing (WES) is a fascinating methodology, examining all ~22,000 known protein-coding genes, which constitute 1–2% of the entire human genome. WES is utilized for genetic testing, particularly in cases of SCD where phenotypes present a wide range of potential diagnoses. It may also be employed as a secondary test when gene panels yield inconclusive results. The diagnostic yield of WES depends on the examined population and the accessibility of familial genetic information, and its complexity lies in evaluating and determining the pathogenic nature of the numerous variants it identifies. Whole-genome sequencing (WGS) analyzes a substantial portion of the entire genome, offering insights into regulatory, intronic, and intergenic regions. Similar to WES, the indications for WGS application are the cases where it is not clear which genes are involved, making it a method to potentially uncover novel or unexpected genetic variants. However, the extensive data output of WGS generates some issues regarding its analysis and storage, which can restrict its practicality. Additionally, WGS typically incurs a higher cost compared to WES or gene panels. Regardless of the method used, whether it is WGS, WES, or gene pane, NGS identifies a large number of genetic variants, many of which fall into the category of undetermined significance (VUS), creating a significant challenge in contemporary genetic investigations. In the ClinVar Miner website, developed by a working group from Utah University [50], out of a total of 12,923 variants identified in the Titin gene (TTN), 7859 variants are classified as VUS [51]. The TTN is recognized as a significant contributor to cardiovascular disease, and 15 to 25% of individuals diagnosed with DCM have been found to harbor a heterozygous TTN-truncating variant as the causative genetic factor [52]. This highlights the substantial necessity to ascertain the pathogenicity of VUS, as misinterpretation of such variants can result in erroneous diagnoses, unnecessary treatments, and significantly influence the mental well-being of individuals [53]. In a familial study performed by Tsai et al., in a 50-family cohort, 61% of VUS were reclassified, 84% of which were classified as B or LB [54]. To enhance the reclassification of VUS, besides conducting genetic testing on family members, recommendations include the development of in vitro tests and predictive biochemical algorithms, complemented by clinical databases [51].

### 4.2. Sample Collection for Molecular Autopsy

According to the Molecular Biology Committee of the American National Association of Medical Examiners, biological samples should be collected from all individuals aged 40 or younger who experience sudden and unexplained deaths [55]. In cases of out-of-hospital cardiac arrest in the young, Stanasiuk et al. recommend the collection of a 2 mL blood sample (in an ethylenediaminetetraacetic acid (EDTA) tube) by the emergency medical service (EMS). Their recent investigation indicates that the isolation yield and DNA integrity from blood samples collected by the EMS seem to surpass those obtained from biological samples collected during autopsy [56]. If blood samples are not collected by the EMS, blood or tissue suitable for DNA extraction should be obtained during the autopsy process. Following the comprehensive autopsy, if there is suspicion of a genetic disease or the underlying cause remains unidentified, it is advisable to extract and store DNA for future analysis [55,57]. As per the Heart Rhythm Society (HRS) and the European Heart Rhythm Association (EHRA) consensus publication on genetic testing for cardiomyopathies and channelopathies, the most appropriate sample to collect at the time of the autopsy for future genetic testing are 5–10 mL of whole blood in a EDTA tube, blood spot card, or a frozen sample of heart, liver, or spleen [43] (Figure 4). According to the recommendations of the Swiss Society of Legal Medicine concerning the collection and storage of appropriate samples for genetic testing, post-mortem material suitable for such testing (such as EDTA blood and/or tissue) should be frozen at a minimum of −20 °C, preferably at −80 °C. These samples should remain in the freezer until they can be sent to a suitable laboratory upon request or discarded in accordance with a retention schedule, with storage for at least 5 years requiring permission from the designated authority [58]. However, Middleton et al. suggest that tailored storage guidelines may be established for individual death investigation offices, especially if they regularly send such testing to a specific laboratory with its own preferred storage conditions [57].

In cases of historical SCD, formalin-fixed and paraffin-embedded tissue (FFPET), typically used for histological analysis, might be the only biological samples readily available. However, the formalin fixation process causes DNA degradation into fragments with an average length of approximately 150 base pairs; chemical modification, such as the deamination of cytosine, which is a significant source of artifactual variations in FFPE samples; and cross-linking between DNA and proteins, which may impact DNA extraction and subsequent enzymatic reactions [59]. Due to DNA fragmentation, the analysis of these samples using Sanger sequencing would be challenging; however, NGS, with its shorter read lengths, can effectively address these limitations [60,61,62,63]. Moreover, the other issues of these samples can be addressed by assessing the quality of extracted DNA, which aids in selecting the optimal DNA extraction and library preparation approaches [59].

### 4.3. Analysis of NGS Data

The analysis of NGS data is a multistep process, and different files are produced through the application of bioinformatic tools: FASTQ files, encompassing base calls for all generated reads and the corresponding quality score of each base; binary aligned/mapped (BAM) files, providing read alignment against the genome of reference; and variant call format (VCF) files, including the chromosomal location, designation, and reference genome for every identified variant. NGS yields a large number of variants that require subsequent filtering and prioritization for pathogenicity evaluation.

#### 4.3.1. Variant Filtration

The procedure of recognizing discrepancies between the reads and the reference genome is termed ‘variant calling’. Following the identification of variants, a filtration step is essential to address potential errors arising from sequencing and alignment inaccuracies. Specialized statistical tools are employed to filter variants based on the probability that an identified mismatch reflects either a technical error or a genuine genetic variant. Genetic variants are typically identified using a quality score, which includes a read coverage of at least 30-fold (alignment of bases to a specific nucleotide position) and a read percentage of 20 or more (indicating the proportion of bases that differ from the reference sequence) [47]. Detecting missense mutations caused by single nucleotide polymorphisms is more easily identified through NGS, while the likelihood of identifying DNA insertions and deletions (indels) decreases as the size of the indel increases. This is primarily due to a higher occurrence of alignment errors [64]. Hereafter, the genetic variants identified must undergo a second filtration step based on their biological aspects, and rare variants are distinguished from those lacking biological relevance in the general population, thereby characterized as background noise. The term signal-to-noise ratio represents the ratio of rare variants in the DNA sample analyzed with background noise [7]. In the initial phases of genomic research, the lack of a specific genetic variation in a group of healthy individuals was regarded as a valid indicator of its potential to cause disease. However, nowadays, the novelty of a mutation is no longer seen as a reliable criterion for interpreting its clinical significance, and different criteria are used in this step of filtering. This includes the frequency of the variant in human genetic databases (such as those with a minor allele frequency MAF less than 0.1%), the gene involved (and its previous disease association), and the type of genetic variant. Missense variants are frequently found in individuals without any apparent health issues, making it challenging to establish a direct connection between these genetic variants and their effects on an individual’s traits. In contrast, nonsense variants, such as deletions, insertions, and splice-site-disrupting variants, are more inclined to lead to the production of abnormal proteins and consequently have a clinical influence. As a result, ‘nonsense’ genetic variants are less common and less likely to be detected in healthy individuals [7].

#### 4.3.2. Variant Prioritization

After the ‘technical’ and ‘biological’ filtration of the variants present in the VCF file, the biological significance of the remaining variants must be analyzed. A previous description is a pivotal criterion that assists in directing the evaluation of its clinical significance. Established guidelines are available with the purpose of standardizing this procedure [65,66]. Public databases of genetic variants, such as ClinVar and OMIM, provide relationships between human genetic variation and observed health status, along with the history of that interpretation. Moreover, computational tools such as Mutation Taster, Polyphen2, FATHMM, DANN, MutationAssessor, and Sift allow the prediction of the impact of a genetic variant on the protein. The conservation status is another important consideration. Specifically, substitutions of amino acids in protein domains that exhibit conservation in other human proteins with similar functions (paralogs) or in the same protein across different species (orthologs) tend to be more significant in a clinical context. Specialized software like GERP++ or PhyloP can evaluate the conservation of DNA sequences within and between species [7,49]. The standards and guidelines for the interpretation of sequence variants developed jointly by the American College of Medical Genetics and Genomics (ACMG) and the Association for Molecular Pathology (AMP) recommend the adoption of standardized terminology for variant classification. This terminology consists of five categories, reflecting a probabilistic attribution to disease association: pathogenic (P), likely pathogenic (LP), likely benign (LB), benign (B), and variant of unknown significance (VUS) [65].

### 4.4. Application of Molecular Autopsy Using NGS

Several recent studies have employed NGS in molecular autopsy [67]; however, direct comparisons of diagnostic rates between different studies are not always feasible due to variations in the genes examined, the number of cases analyzed, and the methodologies employed for prioritizing variants. In general, when compared to molecular autopsy using Sanger sequencing, NGS studies have brought to the forefront the potential involvement of cardiomyopathy and channelopathy genes in some cases of SCD. This association is particularly evident in situations involving subtle cardiac abnormalities that may not be detected during a medico-legal autopsy, and even in cases where such abnormalities are absent.

There are studies that have utilized WES or gene panels containing hundreds of genes in the investigation of genetic variants in victims of sudden cardiac death [2,68,69,70], as there are many genes associated with channelopathies and cardiomyopathies, each with numerous missense, nonsense, insertion/deletion, frameshift, and splice site mutations. Despite the current capability of analyzing large genomic regions through NGS, the 2023 ESC guidelines for the management of cardiomyopathies recommend initially applying genetic testing that focuses on genes strongly associated with the observed phenotype. If initial testing fails to provide a diagnosis but suspicion of a monogenic cause remains high, more comprehensive sequencing or analysis may be appropriate, taking into account family structure and other relevant factors. Once a genetic cause is identified, relatives may undergo testing specifically for the causative variant [71].

A summary of key findings from studies utilizing NGS approaches to investigate cases of sudden death is presented in Table 3.

Girolami et al. [70] analyzed 174 genes to investigate the genetic basis of sudden cardiac arrest (SCA) and SCD in 22 subjects under the age of 50. From the 22 samples analyzed (14 collected at autopsy and 8 from resuscitated patients after SCA), an LP variant associated with cardiomyopathy or channelopathy was identified in four individuals (18%), while 17 (77%) carried a VUS [70]. Hata et al. [67] used NGS to perform molecular autopsies, focusing on 70 genes in 25 individuals who were victims of sudden unexplained death syndrome (SUDS) aged 19–50 years. The results included the identification of 5 known and 15 potentially pathogenic variants in 14 cases (60%). Some of these variants were associated with channelopathies and cardiomyopathy, and a few were linked to pathological changes [67]. Farrugia et al. [72] applied post-mortem genetic testing to 16 individuals who were victims of sudden unexplained death (SUD) aged less than 35 years old. The molecular analyses focused on a set of 23 genes known to be linked with inherited cardiac channelopathies. On average, around 200 variants were identified per case. Yet, following prioritization, four LP variants, encompassing two previously unreported variants, were identified in three cases (18.75%) in the genes KCNH2, ANK2, SCN5A, and RYR2 [72]. In a study by Fadoni et al., a panel encompassing 40 genes was examined in 16 young SCD victims aged between 16 and 50 years, suspected of exhibiting manifestations of HCM. Within this cohort, a pathogenic or likely pathogenic genetic variant was found in 37.5% of victims, predominantly in the most common genes associated with HCM (MYBPC3 and MYH7). One victim (6.25%) presented a VUS suggesting potential pathogenicity, and 10 VUS were identified in nine victims, including an individual with a pre-mortem diagnosis of HCM [73]. Hertz et al. [3] investigated a panel of 100 genes in 72 suspected SCD victims (<50 years). In the post-mortem genetic screening of SCD victims, variants indicating probable functional impacts were identified in 29% of cases with non-diagnostic structural abnormalities of the heart and in 35% of cases with a diagnostic cardiac abnormality [3]. Campuzano et al. [74] studied a cohort of individuals within the same age range as those in the previously mentioned study; however, these 52 individuals were victims of sudden arrhythmic death during exercise. NGS was used to screen 78 genes associated with SCD, and a rare variant classified as deleterious was identified in 53.84% of the victims [74]. Lahrouchi et al. assessed 57 cases of SCD in young individuals with structural abnormalities identified during cardiac autopsy, including 28 cases diagnosed with cardiomyopathy and 29 cases with non-specific findings. They applied an NGS panel consisting of 77 genes associated with primary electrical disorders and cardiomyopathies and identified a P/LP variant in 10 cases (18%). In this study, the majority of identified variants were found in cases diagnosed with cardiomyopathy (9 out of 28 [32%]) compared to those with non-specific findings (1 out of 29 [3%]). These findings suggest that the detection rate of P/LP variants through molecular autopsy in cases involving cardiomyopathy is comparable to that observed in living patients but notably higher than in SCD cases with non-specific structural cardiac abnormalities [75]. Ripoll-Vera et al. used an NGS gene panel containing between 194 and 380 genes to perform a molecular autopsy in 62 individuals who died from nonviolent SCD aged ≤ 50 years. The diagnostic yield of this study was similar to other published studies, as a P/LP genetic variant was identified in 30.6% of the victims [68]. However, the diagnostic yield of NGS applied in post-mortem genetic studies can be much lower than that obtained in the studies mentioned above. Dewar et al. employed a panel comprising 38 candidate genes associated with inherited arrhythmia syndromes, along with an additional 33 genes not previously explored for variants potentially linked to SUD in young individuals. The study focused on investigating 191 cases of child deaths (under 5 years of age) attributed to SUD. The diagnostic yield of the study was 6.3%, revealing 11 potentially causative disease-associated variants in 12 cases. However, 31 VUS were identified in 36 cases, and 16 novel variants predicted to be pathogenic in silico were found in 15 cases [76]. Campuzano et al. [77] explored 108 genes linked to SCD using 44 post-mortem samples from infants aged between 2 and 4 months who passed away while at rest. A definitive cause of death was not established in any of the cases following autopsy. The diagnostic yield of molecular autopsy in this study was 20% of cases, and 63.64% of the victims analyzed in this study carried rare VUS [77]. In 2014, Bagnall et al. [2] conducted WES for the first time in 28 young individuals who were victims of SUD. They identified three rare variants in LQTS-associated genes. The broadening of the analysis to include other genes linked to channelopathies and cardiomyopathies resulted in the additional identification of six rare variants [2]. In a subsequent investigation, the same research team conducted gene panel analysis on 51 sudden unexplained death (SUD) cases, which included either 69, 98, or 101 genes, and performed WES, focusing on 59 cardiac-related genes in another 62 SUD cases. This analysis revealed the presence of a genetic variant in a cardiac-related gene of clinical significance in 31 cases, accounting for 27% of the cases [1]. Anderson et al. [78] evaluated the role of WES in exertion-related SUD cases involving 21 young individuals who had previously undergone a molecular autopsy, focusing on four major channelopathy-susceptibility genes (KCNQ1, KCNH2, SCN5A, and RYR2). The application of WES in this cohort focused on 100 genes, and pathogenic genetic variants were identified in three of them (two individuals had a P variant in CALM2, and one in the PKP2). Among the remaining 18 cases, seven harbored at least one VUS with a minor allele frequency of less than 1 in 20,000 [78]. Nunn et al. [79] performed a molecular autopsy in 59 victims of SADS (aged 1–51 years), applying WES focusing on 135 genes associated with cardiomyopathies and ion channelopathies. A P/LP genetic variant was identified in 17 probands (28.81%), and very rare variants (<0.02% in NHLBI and UCL-exomes cohorts) were identified in seven probands in the genes SCN5A, TTN, RyR2, GJA5, MYOT, and DSC2. VUS were detected in 34% of probands. Many of these VUS will be plausible candidate P/LP genetic variants, and the challenge is to then determine their pathogenicity [79]. Neubauer et al. applied exome analysis to investigate a cohort of 34 unexplained death (SUD) cases by focusing on 192 genes of interest (candidate genes associated with channelopathies and cardiomyopathies). In this SUD cohort, potentially disease-causing genetic variants were identified in 29.4% of the 34 SUD cases, and variants with likely disease-causing effects were identified in ACAD9, AKAP9, BMPR2, EFEMP2, FBN2, KCNE5, MYLK, and RYR2, among others [80]. Recently, Neves et al. [69] applied WES in 38 unexplained SCA survivors (age 1–56 years) and 68 autopsy-inconclusive SUD cases (average age at SCD of 20.4 ± 9.0 years). Utilizing the ACMG criteria for variant classification, a P/LP variant was identified in 3 out of 38 (7.9%) SCA survivors and 8 out of 68 (11.8%) SCD cases with inconclusive autopsy findings [69]. The studies mentioned in the provided text collectively advance our knowledge of the role of molecular autopsies in understanding sudden cardiac death (SCD), particularly in young individuals. From pioneering efforts like Ackerman et al.’s early use of Sanger sequencing to more recent studies utilizing next-generation sequencing technologies, these investigations highlight the importance of genetic testing in unexplained cases of SCD. They emphasize that a significant proportion of SCD cases may be underpinned by genetic factors, underscoring the value of genetic screening in both diagnostic and forensic contexts. These studies contribute to ongoing efforts to uncover the genetic causes of SCD, offering hope for improved prevention, diagnosis, and management of these devastating events. It is important to note that the data presented in these studies show that there seems to be no positive correlation between the expansion of the genomic regions investigated and the number of conclusive genetic tests, and the large amount of data produced by WES increases the number of VUS identified. Therefore, the diagnostic yield of genetic testing using NGS does not depend solely on the number of genomic regions analyzed but also on the ability to interpret the disease association status of the VUS identified [82]. Comparing the pros and cons of the NGS approaches available, the targeted sequencing using gene panels seems to be the most appropriate for this research, with WES being applied in cases where it is not clear which genes are involved.

#### Molecular Autopsy Using FFPE Samples

In 2017, Baudhuin et al. were the pioneers in utilizing formalin-fixed paraffin-embedded tissue (FFPE) samples for genomic analysis in four cases exhibiting clinical features indicative of an inherited cardiovascular disorder [60]. In the same year, Bagnall et al. were the pioneers in demonstrating the feasibility of conducting NGS on FFPE samples from cases of juvenile SCD [61]. In a recent study, a comparison was made between the results of NGS analysis on 12 sudden cardiac death cases using FFPET samples and their corresponding non-formalin-fixed samples (such as RNA-later-preserved tissues or bloodstain cards). It was observed that all P variants, LP variants, or VUS identified in the non-fixed samples were also validated in the FFPE samples, albeit with varying levels of confidence. However, the FFPE samples exhibited more false positives and negatives, especially when subjected to formalin fixation for a period longer than 8 days [62]. In 2021, van der Crabben et al. [63] used DNA isolated from paraffin-embedded tissue from two SCD female victims and used NGS to analyze a panel of 56 cardiomyopathy-associated genes in a 60-year-old victim, in which a pathogenic variant was identified in the FLNC gene, and 54 arrhythmia-associated genes were investigated in a 41-year-old victim, in which a genetic variant classified as pathogenic was identified in the KCNH2 gene, associated with long QT syndrome type 2 (LQTS 2). Moreover, in this study, the above-mentioned 56 cardiomyopathy panel was used to perform a molecular autopsy on a 33-year-old male victim of SCD. In this case, DNA isolated from material derived from an autopsy revealed a pathogenic genetic variant in the FLNC gene [63].

Modena et al. conducted WES on 32 post-mortem samples from SCD victims aged 50 years or younger. The analyzed samples comprised 11 FFPE tissues from the heart, kidney, or spleen, and 21 blood samples collected in EDTA tubes. WES was refined using a customized virtual gene panel, followed by structured variant prioritization and multidisciplinary evaluation. This approach identified a likely causative variant in 69% (22 of 32 cases), highlighting the effectiveness of WES in post-mortem genetic analysis, including FFPE samples (Modena, 2024) [81]. However, one FFPE sample was not included in the analysis due to bacterial contamination, which likely compromised DNA integrity and the quality of reads, hindering the accurate identification of genetic variants. Besides the issue of bacterial contamination, FFPE samples present additional challenges that NGS can address but are difficult for Sanger sequencing to overcome. These studies demonstrate that NGS, particularly suited for the analysis of degraded DNA due to its ability to sequence shorter fragments, allows for reliable genetic data even from compromised samples.

### 4.5. Limitations of NGS in Molecular Autopsies

The advent of NGS has unquestionably transformed the application of molecular autopsies. However, like any powerful tool, NGS is not without its limitations. Its application in the forensic routine is influenced by various factors, including cost, accessibility, and resource availability. The cost of a molecular autopsy can vary based on factors such as the number of genes analyzed, the specific methodology and platforms used, and regional pricing differences. The estimated cost per analysis is estimated to vary between USD 240–297 for targeted panels, USD 604–1932 for WES, and USD 2006–3347 for WGS. Cost proportions are highest for library preparation and sequencing materials (average 76.8% of total costs), sample extraction (8.1%), data analysis (9.2%), and data storage (2.6%). Capital costs for the sequencers are an additional USD 24–67 per person [83].

Regarding technical aspects, NGS does not provide uniform precision when characterizing all segments of the genome. For instance, genomic regions abundant in cytosine and guanine nucleotides pose sequencing challenges due to the stronger bonds between DNA strands, which restrict their accessibility during the replication process. As coverage decreases in these regions, variant calls within them may go unassessed. NGS is also susceptible to alignment errors, primarily affecting areas with InDels, as well as regions in the DNA featuring extended repeated sequences beyond the length of the short reads. These factors collectively constitute potential sources of false negative results in molecular autopsy of SCD victims [49,64]. Given the intricate nature of bioinformatic analysis associated with NGS, the ongoing advancements in the field, and the significant impact of assigning pathogenicity to a particular variant, the process of translating genetic data into a clinical context necessitates specialized expertise and continuous training. With the expansion of the application of NGS gene testing, an increase in the detection of VUS ensues. Presently, this stands as the primary limitation of NGS molecular autopsy, as these VUS cannot be correlated as the causal genetic variant that leads to SCD. Clarifying the diagnostic significance of a VUS, shifting it to either LP/P or LB/B, is key to increasing the diagnostic yield of molecular autopsies. Continual advancements in genetic data concerning rare variants may lead to revisions of prior classifications, highlighting the importance of regularly reassessing the classification of genetic variants. Recently, Campuzano et al. conducted a re-evaluation of a cohort comprising 104 subjects diagnosed with inherited arrhythmogenic syndromes and 17 post-mortem cases where inherited arrhythmogenic syndromes were identified as the cause of death. This study identified that 71.87% of rare variants associated with these conditions had undergone a change in their classification [84]. This variant reinterpretation is primarily prompted by a clinician’s request, the identification of a previously classified variant in a new patient, or the availability of new data related to the rare variant [85]. Vallverdú-Prats et al. [86] evaluated 39 rare variants associated with ACM, initially classified in 2016 according to the ACMG guidelines. Five years later, the team re-evaluated these rare variants using updated data. After conducting a thorough reanalysis, 30.77% of the variants underwent a classification change, primarily attributed to the incorporation of updated global frequencies. While 64.1% of variants in 2016 were deemed to have an uncertain role, this proportion notably decreased to 17.95%, and the percentage of rare variants considered potentially deleterious rose from 17.95% to 23.07%. Notably, 83.33% of reclassified variants gained certainty [86]. Nonetheless, as NGS becomes more widely adopted and as more data accumulates regarding variants linked to SCD, along with the advancement of more accurate tools for predicting the impact of individual or combined genetic variants, the significance of many VUS may be clarified in the near future [49,84].

## 5. Discussion

The focus of this review is to investigate the applications of NGS in the identification of genetic variants associated with SCD, especially when a medico-legal autopsy is unable to determine the cause of death. NGS has revolutionized genetic analysis by enabling high-throughput sequencing of large regions of the human genome in a time-efficient way [11]. Compared to Sanger sequencing, the ability of NGS to sequence millions of DNA fragments simultaneously has drastically advanced molecular investigations. This ability is especially important in SCD cases with a genetic basis, such as cardiomyopathies and channelopathies, associated with a vast number of genetic variants. The integration of NGS into the molecular autopsy workflow represents a paradigm shift in the post-mortem analysis of SCD.

The studies analyzed in this review showed that, in comparison to broader approaches like WES or WGS, targeted gene panels that focus on genes strongly associated with specific phenotypes achieve higher diagnostic yields. The studies here analyzed achieved diagnostic yields ranging from 3% to 60%, depending on the cohort and methodology applied.

One of the most critical challenges in molecular autopsy using NGS is the interpretation of VUS. These variants lack sufficient evidence to be classified as benign or pathogenic; thus, the uncertainty surrounding VUS is a challenge in the identification of a definitive cause of death. Published studies suggest the reassessment of VUS, as these variants may be reclassified over time based on developments in public databases, computational prediction tools, and functional studies [84,85,86]. Recent advancements in multi-omics integration and bioinformatics tools have significantly improved the interpretation of genetic data. Multi-omics, which combines various omics fields including genomics, transcriptomics, proteomics, and metabolomics, provides a more comprehensive view of how genetic variants impact biological pathways and disease mechanisms [87], including sex-specific differences in cardiac pathologies [88]. Additionally, advanced bioinformatics tools, such as machine learning algorithms and artificial intelligence-driven predictive models, are enhancing variant classification by integrating vast amounts of genetic, clinical, and population-level data [89]. Databases like ClinVar and gnomAD, coupled with computational tools such as REVEL (rare exome variant ensemble learner, an ensemble method for predicting the pathogenicity of missense variants on the basis of the individual tools MutPred, FATHMM, VEST, PolyPhen, SIFT, PROVEAN, MutationAssessor, MutationTaster, LRT, GERP, SiPhy, phyloP, and phastCons) [90], facilitate more precise pathogenicity predictions. The integration of these innovative approaches into molecular autopsy not only improves diagnostic accuracy but also provides critical insights for risk assessment and preventive strategies in surviving relatives of SCD victims.

Another challenge to the application of NGS in molecular autopsies is the interpretation of regions with high guanine–cytosine content, large indels, and repetitive sequences. To ensure accuracy, the sequencing results of these regions need to be carefully validated [49,64].

The Molecular Biology Committee of the American National Association of Medical Examiners recommends the collection of a biological sample from all individuals aged 40 years or younger who are victims of sudden and unexplained deaths. If blood samples in EDTA tubes are not collected by the EMS, blood or tissue suitable for DNA extraction should be obtained during the autopsy process. According to the Heart Rhythm Society and the European Heart Rhythm Association, the most appropriate sample to collect at the time of the autopsy for future genetic testing is 5–10 mL of whole blood in an EDTA tube, blood spot card, or a frozen sample of heart, liver, or spleen [43].

In cases of historical SCD, FFPE samples might be the only available biological sample for molecular autopsy; however, the formalin fixation process can lead to DNA degradation [40]. The ability of NGS to analyze shorter DNA fragments has proven to be effective in overcoming this limitation, as studies included in this review demonstrate the feasibility of NGS in identifying pathogenic variants using FFPE samples for NGS [62,63,81]. However, challenges such as false positives and false negatives highlight the importance of optimized protocols and validation of techniques [62].

The utility of NGS extends beyond the identification of the genetic causes of SCD. By elucidating pathogenic variants in SCD victims, molecular autopsy provides critical insights for the genetic screening of first-degree relatives.

In cases where a molecular autopsy identifies a genetic predisposition to sudden arrhythmic death syndrome (SADS), implementing preventive strategies for at-risk family members is crucial. One primary intervention is the use of defibrillator devices, which monitor heart rhythms and deliver shocks to correct life-threatening arrhythmias. The main types of defibrillators include implantable cardioverter–defibrillators (ICDs) and wearable cardioverter–defibrillators (WCDs), which are powerful tools in preventing SCD due to ventricular arrhythmias in carefully selected patients [91].

The integration of molecular autopsy into the forensic workflows also has broader implications for public health. It offers the potential to establish genetic markers that can guide preventive measures at a population level, particularly in cases where SCD occurs without prior symptoms.

It is expected that interpretation of large-scale NGS data, the accuracy of bioinformatics analyses and variants classification can be improved by the advancements in artificial intelligence, machine learning the integration of multi-omic approaches.

## 6. Conclusions

The rapid advancement and integration of NGS with its high-throughput screening capabilities have substantiated additional genes and genetic variants associated with hereditary cardiovascular diseases. This expansion of the genetic spectrum contributes to a broader understanding of the molecular etiology of SCD, reinforcing the importance of post-mortem genetic analysis. A crucial factor in ensuring the success of molecular autopsy is the proper collection of biological samples, particularly in individuals aged 40 or younger who experience sudden and unexplained deaths. EMS should collect a 2 mL EDTA blood sample whenever possible. If unavailable, blood or tissue should be obtained during autopsy to ensure high-quality DNA for genetic analysis. This practice enhances diagnostic accuracy, facilitates family risk assessment, and supports prevention strategies.

In summary, as sequencing technology and molecular biology research have advanced, there has been a gradual understanding of the genetic mechanisms underlying SCD. However, to further enhance the clinical utility of NGS in forensic investigations, actionable strategies should be implemented. First, there is a need to standardize gene panels for specific SCD phenotypes to ensure consistent and comprehensive analysis of relevant genetic variants across different cases. This would allow for more reliable and reproducible investigation methodologies when analyzing post-mortem samples. Second, improving the interpretation of VUS could contribute to the accurate classification of these variants, increasing the yield of post-mortem molecular testing and reducing ambiguity in clinical decision-making, as the identification of pathogenic genetic variants in SCD victims can not only lead to the clarification of the cause of death but also identify molecular markers to be investigated in the first relatives of the victims, who may also be at risk. In the future, a more profound comprehension of the molecular mechanisms involved in SCD is expected to bring about more effective choices of preventive, diagnostic, and therapeutic methods for inherited cardiovascular diseases that lead to SCD.

## Figures and Tables

**Figure 1 diagnostics-15-00460-f001:**
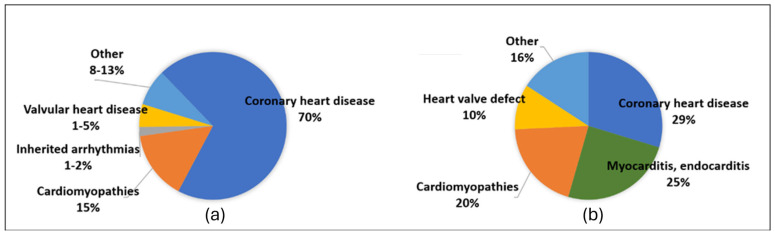
Causes of SCD. (**a**) Causes of SCD across all ages; adapted from Wong et al. [6]. (**b**) Causes of SCD in young adults; adapted from Markwerth et al. [20].

**Figure 2 diagnostics-15-00460-f002:**
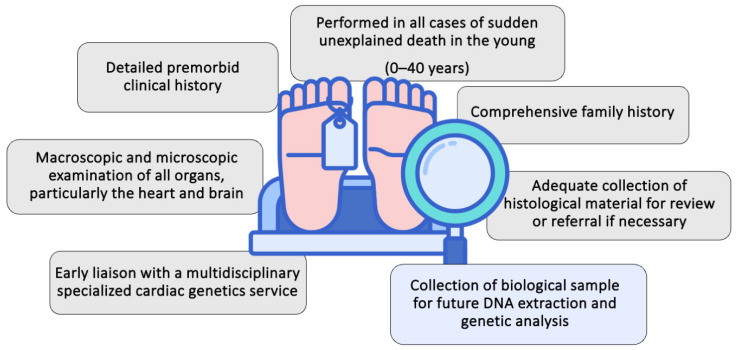
Post-mortem investigation process in sudden cardiac death cases; adapted from Semsarian and Ingles [44].

**Figure 3 diagnostics-15-00460-f003:**
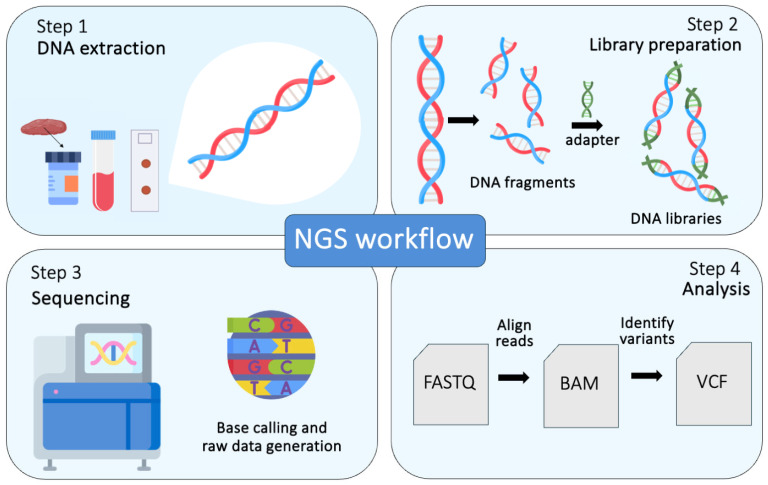
An overview of the steps involved in genetic variant identification using next-generation sequencing. FASTQ file: a text file that contains a numeric quality score associated with each nucleotide in a sequence. BAM file: binary alignment map. VCF file: variant calling format.

**Figure 4 diagnostics-15-00460-f004:**
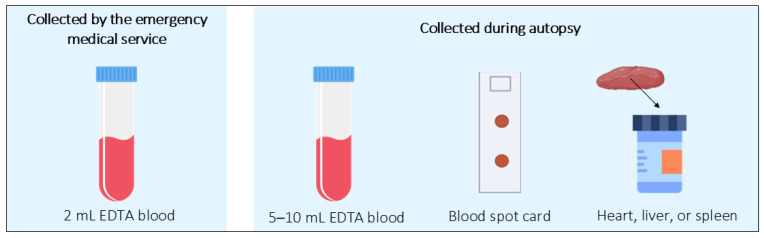
Post-mortem collected samples used in molecular autopsy.

**Table 1 diagnostics-15-00460-t001:** Prevalence, associated genes, and diagnostic yield of genetic testing of cardiomyopathies with robust genetic connections. HCM: hypertrophic cardiomyopathy, DCM: dilated cardiomyopathy, ACM: arrhythmogenic cardiomyopathy.

Genetic Cardiomyopathy	Estimated Prevalence	Associated Genes	Diagnostic Yield of Genetic Testing
HCM	1:200–1:500 [22,23]	ACTC1, ACTN2, ALPK3, ANKRD1, BAG3, CAV3, CSRP3, FHL1, FLNC, JPH2, LDB3, MYBPC3, MYH6, MYH7, MYL2, MYL3, MYLK2, MYOM1, MYOZ2, NEXN, PLN, RYR2, TCAP, TNNC1, TNNI3, TNNT2, TPM1, TTN, VCL [24,25]	30–60% [26]
DCM	1: 250 [23]	ABCC9, ACTC, ACTN2, BAG3, CRYAB, CSRP3, DES, DMD, DSG2, DSP, EMD, EYA4, FLNC, JUP, LAMA4, LBD3, LMNA, MYBPC3, MYH6, MYH7, NEBL, NEXN, PKP2, PLN, RBM20, SCN5A, SGCA, SGCB, SGCD, SGCG, TAZ, TCAP, TNNCI, TNNI3, TNNT2, TPM1, TTN, VCL [25,27]	24–55% [28,29]
ACM	1:200–1:5000 [23]	DES, DSC2, DSG2, DSP, JUP, PKP2, PLN, TMEM43 [25,30,31]	20% [32]

**Table 2 diagnostics-15-00460-t002:** Prevalence, associated genes, and diagnostic yield of genetic testing of genetic channelopathies. LQTS: long QT syndrome, BrS: Brugada syndrome, CPVT: catecholaminergic polymorphic ventricular tachycardia, SQTS: short QT syndrome; * evidence of causality is limited.

Genetic Channelopathy	Estimated Prevalence	Associated Genes
LQTS	1:2000 [34]	AKAP9 *, ANK2 *, CACNA1C, CALM1, CALM2, CALM3, CAV3, KCNE1, KCNE2 *, KCNH2, KCNJ2, KCNJ5 *, KCNQ1, NOS1AP, RYR2, SCN4B *, SCN5A, SLC8A1, SNTA1 *, TERCL, TRDN [10,35]
BrS	1:2000 [36]	ABCC9, CACNA1C, CACNB2, CACNA2D1, FGF12B, GPD1L, GSTM3, HCN4, HEY2, KCND3, KCNE3, KCNE5, KCNAB2, KCND2, KCNH2, KCNJ8, LRRC10, MAPRE2, PKP2, RANGRF, RRAD, SCN5A, SCN10A, SCN1B, SCN2B, SCN3B, SEMA3A, SLMAP, TRPM4, TBX5, XIRP1, XIRP2, ZFHX3 [10,37]
CPVT	1:10,000 [38]	ANK2, CALM1, CALM2, CALM3, CASQ2, KCNJ2, KCNQ1, NKYRIN-B, KCNE1, KCNE2, KCNH2, PkP2, JUN, RyR2, SCN5A, TECRL, TRDN [10,39]
SQTS	1:1000–1:5000 [40]	CACNA1C *, CACNB2 *, CACNA2D1 *, KCNH2, KCNQ1, KCNJ1, KCNJ2, SCN5A, SLC4A3 [10,41,42]

**Table 3 diagnostics-15-00460-t003:** Studies utilizing NGS approaches in sudden death investigation—summary of key findings. SD: sudden death, SCD: sudden cardiac death, SIDS: sudden infant death syndrome, SUD: sudden unexpected death, SCA: sudden cardiac arrest, SADS: sudden arrhythmic death syndrome, HCM: hypertrophic cardiomyopathy.

NGS Approach	N	Age	SD Type	Diagnostic Yield	Reference
174 genes	22	<50 years	SCD and SCA	18%	[70]
70 genes	25	19–50 years	SUDS	60%	[67]
23 genes	16	<35 years	SUD	18.75%	[72]
40 genes	16	16–50 years	SCD (HCM)	37.5%	[73]
100 genes	72	<50 years	SCD	29% with non-diagnostic	[3]
				35% with cardiac abnormalities	
78 genes	52	≤50 years	SADS	53.84%	[74]
77 genes	57	21–38 years	SCD	32% cases diagnosed with cardiomyopathy	[75]
				3% non-specific findings	
194–380 genes	62	≤50 years	SCD	30.6%	[68]
71 genes	191	<50 years	SUD	6.3%	[76]
108 genes	44	2–4 months	SIDS	20%	[77]
69, 98, or 101 genes	51	1–35 years	SUD	27%	[1]
WES focused on 59 genes	62				
WES focused on 100 genes	21	2–19 years	SUD	14%	[78]
WES focused on 135 genes	59	1–51 years	SADS	28.8%	[79]
WES focused on 192 genes	34	1–63 years	SUD	29.4%	[80]
WES	38	1–56 years	SCA	7.9%	[69]
	68	average age of 20.4 ± 9.0 years	SUD	11.8%	
WES focused on 1304 genes	32	≤50 years	SCD	69%	[81]

## Data Availability

No new data were created or analyzed in this study.

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
