# Peer review of "Sudden Cardiac Death: The Role of Molecular Autopsy with Next-Generation Sequencing"

_diagnostics, 2025, doi:10.3390/diagnostics15040460_

Round 1
Reviewer 1 Report
Comments and Suggestions for Authors
Dear Author(s),
I have carefully reviewed your paper titled "Sudden cardiac death: the role of molecular autopsy with next-generation sequencing". The paper addresses an important topic and handles the advancements in next-generation sequencing (NGS) applied to sudden cardiac death (SCD). Below, I listed comments and suggestions aiming the improvement of paper.
The abstract could provide a more balanced overview by briefly acknowledging both the advancements and challenges of NGS in molecular autopsies.
The narrative review format lacks formal criteria for study selection, which may limit reproducibility. I suggest using a systematic review methodology, clearly documenting search strategies, inclusion/exclusion criteria, and databases utilized. Including a PRISMA flowchart would enhance transparency, as well.
The recommendations are general and lack specificity. Proposing actionable strategies, such as standardizing gene panels for specific SCD phenotypes or improving VUS interpretation through collaborative multi-center studies, would strengthen the conclusion.
The manuscript could benefit from greater emphasis on innovative approaches for variant reclassification, such as multi-omic integration and advanced bioinformatics tools.
Adding infographics illustrating NGS workflows or trends in diagnostic yield could add paper’s impact.
A postmortem case management flowchart or algorithm will be helpful in practice of readers. While tables summarizing gene prevalence and diagnostic yield are informative, the addition of schematic diagrams (e.g., depicting the molecular autopsy workflow) would increase understanding of readers.
Sincerely,
Author Response
Thank you for your valuable feedback and insightful comments.
We have carefully addressed all the points raised in the attachment. Please let us know if any further clarifications are needed.

Reviewer 2 Report
Comments and Suggestions for Authors
First of all congratulations for Authors. The subject of sudden cardiac death (SCD) constitutes a major clinical and public health problem, whose death burden is estimated for four million SCDs worldwide every year. The review of this paper presents accomplishments of the molecular autopsy. I appreciate, it was real pleasure and an honour.
Specific comments to authors :
1. What is the cost of molecular autopsy? Please include this information in discussion section.
2. In up to 30% of SCD cases in young individuals, no cause of death is identified post-mortem, referred to as autopsy-negative or sudden arrhythmic death syndrome (SADS). In instances where the cause of death remains unknown, post-mortem genetic testing, also known as molecular autopsy, may identify a cause of in up to 30% of SADS cases . On the basis of the above, in my opinion it would be usufull for the manuscript to include some relevant information for educational purpose for the readers, let’s hope wide medical community, concerning treatment options: ICD/SICD/VEST defibrillator.
3. I truly recommend this paper for publication in in the present form after just a minor revision, with few sentences added in discussion section mentioned above, if the authors agree to do so.
4. I recommend this paper for publication because every new data and diagnostic perspectives in SCD prevention is extremely important. The presented study is easy to read, interesting and extremely helpful in understanding genetic sequencing techniques, such as next-generation sequencing (NGS). This methodology gives hope for relatives of SCD victims, especialy in young SCD cases.
5. Biological samples should be collected from all individuals aged 40 or younger who experience sudden and unexplained deaths. In cases of out-of-hospital cardiac arrest in the young, Stanasiuk et al. recommended the collection of a 2 mL blood sample (in an ethylenediaminetetraacetic acid - EDTA tube) by the emergency medical service (EMS). If blood samples in EDTA tubes are not collected by the EMS, blood or tissue suitable for DNA extraction should be obtained during the autopsy process. This statement is so important and should be emphasized. I agree it should be repeated in the text but my advice is to transfer this from discussion section to conclusions.
Author Response

(The authors gave the same response as above.)

Reviewer 3 Report
Comments and Suggestions for Authors
The article may be useful to the scientific community, but it certainly requires an improvement in clarity and, at the very least, an integration of the analyzed literature.
•It is recommended to streamline the introduction paragraph by moving the discussion on advancements in genetic analysis methods and the differences between them (Sanger vs. NGS) to a dedicated paragraph for further elaboration.
•From your review work, it seems that the following article, which is certainly related to your study, has been overlooked:
DOI: 10.1186/s40246-024-00657-x
Author Response

(The authors gave the same response as above.)

Round 2
Reviewer 3 Report
Comments and Suggestions for Authors
Having reviewed the integration carried out, I believe that the article, as currently presented, can contribute to the knowledge of the scientific community.
Author Response
Comment: Having reviewed the integration carried out, I believe that the article, as currently presented, can contribute to the knowledge of the scientific community.
Response: Thank you very much for your thoughtful feedback. We are grateful for your positive assessment of the article. We are pleased to hear that you believe it can contribute to the knowledge of the scientific community.
Thank you again for your time and consideration.